# TopoPoint: Enhance Topology Reasoning via Endpoint Detection in Autonomous Driving

**Yanping Fu[1,2,3], Xinyuan Liu[1,2], Tianyu Li[3,4], Yike Ma[1], Yucheng Zhang[1], Feng Dai[1]**[*]
[1]Institute of Computing Technology, Chinese Academy of Science;
[2]University of Chinese Academy of Sciences; [3]Shanghai AI Lab; [4]Shanghai Innovation Institute
fuyanping23s@ict.ac.cn

## Abstract

Topology reasoning, which unifies perception and structured reasoning, plays a vital role in understanding intersections for autonomous driving. However, its performance heavily relies on the accuracy of lane detection, particularly at connected lane endpoints. Existing methods often suffer from lane endpoints deviation, leading to incorrect topology construction. To address this issue, we propose TopoPoint, a novel framework that explicitly detects lane endpoints and jointly reasons over endpoints and lanes for robust topology reasoning. During training, we independently initialize point and lane query, and proposed Point-Lane Merge Self-Attention to enhance global context sharing through incorporating geometric distances between points and lanes as an attention mask . We further design Point-Lane Graph Convolutional Network to enable mutual feature aggregation between point and lane query. During inference, we introduce Point-Lane Geometry Matching algorithm that computes distances between detected points and lanes to refine lane endpoints, effectively mitigating endpoint deviation. Extensive experiments on the OpenLane-V2 benchmark demonstrate that TopoPoint achieves state-of-the-art performance in topology reasoning (48.8 on OLS). Additionally, we propose $DET_p$ to evaluate endpoint detection, under which our method significantly outperforms existing approaches (52.6 v.s. 45.2 on $DET_p$). The code is released at https://github.com/Franpin/TopoPoint.

## 1 Introduction

In autonomous driving scenarios, perceiving lane markings and traffic elements on the road surface is critical for understanding complex intersection environments. To enable accurate interpretation of the scene and determine feasible driving directions, it is essential to infer both lane-lane topology and lane-traffic element topology. With the growing trend of end-to-end autonomous driving systems[1, 2, 3], perception and reasoning have become increasingly integrated into a unified task, referred to as topology reasoning[4, 5, 6, 7, 8]. This task also plays a vital role in high-definition (HD) map learning[9, 10, 11, 12] and supports downstream modules such as planning and control.

As a continuation of the lane detection task, topology reasoning task need to uniformly process lanes, traffic elements, and their corresponding topological relationships, so the query-based architecture has become the mainstream solution. In this pipeline, the multiple lanes are encoded and predicted through multiple independent queries, as shown in Figure 1(a). However, since the lane endpoints are actually attached to lane query and are affected by the supervised learning of multiple lanes, it is difficult to ensure that the multiple endpoints of the final prediction can strictly coincide, which is called the *endpoint deviation* problem. This problem already explored preliminarily as early as in the era of lane detection, e.g., the method STSU[13] aligns the endpoints by moving the entire lane, while the method LaneGAP[14] adopts a path-wise modeling approach, predicting complete

---

[*]Corresponding Author

39th Conference on Neural Information Processing Systems (NeurIPS 2025).

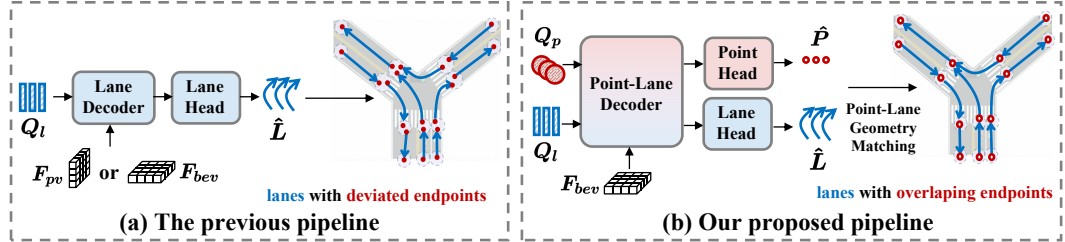

Figure 1: **Pipeline Comparison.** (a) In the previous pipeline, lanes are predicted independently, which leads to obvious endpoint deviation. (b) In our proposed pipeline, lane endpoints are explicitly modeled, and lanes with overlapping endpoints are obtained through point-lane geometry matching.

lane paths by merging connected lane pieces. However, due to the suboptimal performance of lane detection, these methods have been replaced. A recent work, TopoLogic[15], has once again noticed this problem. It integrates the lane-lane geometric distance and semantic similarity to alleviate the interference of the endpoint deviation in topology reasoning, instead of rectifying the issue itself. Therefore, lane detection is still inaccurate, which means that the endpoint deviation problem has not been completely resolved.

To address the aforementioned issues, we propose TopoPoint, a novel framework that introduces explicit endpoint detection and fuses features from both lanes and endpoints to enhance topology reasoning, as is illustrated in Figure 1(b). By reasoning over the topological relationship between endpoints and lanes, TopoPoint effectively mitigates the endpoint deviation problem. To enable point detection and facilitate feature interaction between points and lanes during training, we design the point-lane detector, independently initializing point query and lane query. These queries are supervised at the output by separate objectives for lane detection and endpoint detection. We further propose Point-Lane Merge Self-Attention (PLMSA), and it concatenates point and lane query and leverages geometric distances as attention masks to enhance global context sharing. To enhance point-lane feature interactions, we introduce the Point-Lane Graph Convolutional Network (PLGCN), and it models the topological relationships between points and lanes by constructing an adjacency matrix. This enables bidirectional message passing between point and lane features through Graph Convolutional Network (GCN)[16]. PLGCN serves as a key component of our Unified Scene Graph Network. This joint learning process significantly enhances the representation capability of both endpoints, lanes and traffic elements, thereby improving topology reasoning performance. During inference, we propose the Point-Lane Geometry Matching (PLGM) algorithm, and it computes geometric distances between detected endpoints and the start and end points of lanes. This allows us to refine lane endpoints by matching points to lanes based on their geometric proximity, effectively mitigating the endpoint deviation issue. Our contributions are summarized as follows:

1. We identify that the endpoint eviation issue in current methods stems from the fact that lane endpoints are simultaneously supervised by multiple lanes. To tackle this, we propose independently detecting endpoints and Point-Lane Geometry Matching algorithm to refine lane endpoints.

2. We introduce TopoPoint, a novel framework designed to enhance topology reasoning by incorporating explicit endpoint detection. Within TopoPoint, point query and lane query exchange global contextual information through the proposed Point-Lane Merge Self-Attention, and their feature interaction is further reinforced by the Point-Lane Graph Convolutional Network.

3. All experiments are conducted on the OpenLane-V2[17] benchmark, where our method outperforms existing approaches and achieves state-of-the-art performance. In addition, We introduce $DET_p$ for evaluating endpoint detection, and our method achieves notable improvements.

## 2 Related Work

### 2.1 Lane Detection

Lane detection is essential for autonomous driving, providing structural cues for road perception[9, 12, 11, 10] and motion planning[3]. Traditional methods typically use semantic segmentation to identify lane areas in front-view images, but they often struggle with long-range consistency and occlusions.

To overcome these limitations, vector-based approaches model lanes as sparse representations. Recent advances in 3D lane detection have been driven by sparse BEV-based object detectors like DETR3D[18] and PETR[19], which use sparse query and multi-view geometry to reason directly in 3D space. These ideas have inspired a new wave of lane detectors. For instance, CurveFormer[20] represents lanes with 3D line anchors and introduces curve query that encode strong positional priors. Anchor3DLane[21] extends LaneATT[22]'s line anchor pooling and incorporates both intrinsic and extrinsic camera parameters to accurately project 3D anchor points onto front-view feature maps. PersFormer[23] leverages deformable attention to learn the transformation from front-view to BEV space, improving spatial alignment. LATR[24] further refines lane modeling by decomposing it into dynamic point-level and lane-level query, enabling finer topological representation.

## 2.2 Topology Reasoning

Topology reasoning in autonomous driving aims to interpret road scenes and define drivable routes. STSU[13] encodes lane query for topology prediction by DETR[25]. LaneGAP[14] applies shortest path algorithms to transform lane-lane topology into overlapping paths. TopoNet[26] combines Deformable DETR[27] with GNN[28] to aggregate features from connected lanes. TopoMLP[29, 30] leverages PETR[19] for lane detection and uses a multi-layer perceptron for topology reasoning. TopoLogic[15] integrates geometric and semantic information by combining lane-lane geometric distance with semantic similarity. TopoFormer[31] introduces unified traffic scene graph to explicitly model lanes. SMERF[32] improves lane detection by incorporating SDMap as an additional input, while LaneSegNet[33] uses Lane Attention to identify lane segments. In our work, We introduce endpoint detection to enhance topology reasoning and mitigate endpoint deviation.

## 3 Method

### 3.1 Problem Definition

Given surround-view images captured by multiple cameras mounted on a vehicle, the topology reasoning task includes: 3D lane centerline detection[34, 19, 35, 36, 23] in the bird's-eye view (BEV) space, 2D traffic element detection[37] in the front-view image, topology reasoning[26, 33, 17, 32] among lane centerlines and topology reasoning between lane centerlines and traffic elements. All lane centerlines are represented by multiple sets of ordered point sequences $L = \{l_i \in \mathbb{R}^{k \times 3} | i = 1, 2, \ldots, n_l\}$, where $n_l$ is the number of lane centerlines and $k$ is the number of points on the lane centerline. All traffic elements are represented using multiple 2D bounding boxes $T = \{t_i \in \mathbb{R}^4 | i = 1, 2, \ldots, n_t\}$, where $n_t$ is the number of traffic elements. The lane-lane topology, which encodes the connectivity between lanes, is represented by an adjacency matrix $G_{ll}$. The lane-traffic element topology, capturing the association between lanes and traffic elements, is represented by another adjacency matrix $G_{lt}$. In addition, the framework includes point detection and point-lane topology reasoning. A set of candidate points $P = \{p_i \in \mathbb{R}^3 | i = 0, 1, 2, \ldots n_p\}$ is constructed by de-duplicating all endpoints of lane centerlines, where $n_p$ is the number of unique endpoints. The point-lane topology $G_{pl}$ is created by checking whether the point lies on lane centerline.

### 3.2 Overview

As illustrated in Figure 2, our proposed TopoPoint framework consists of traffic detector, point-lane detector, geometric attention bias, topology head and point-lane result fusion. We downsample the multi-view by a factor of 0.5, while keeping the front-view at its original resolution. During training, all images are passed through ResNet-50[38] pretrained on ImageNet[39] with FPN[40] to extract multi-scale features. These features are then encoded into BEV representations using BevFormer[41] encoder. In the traffic detector, front-view features are directly processed by Deformable DETR[27] to produce traffic query $\hat{Q}_t$. In the point-lane detector, point query $Q_p$ and lane query $Q_l$ interact via Point-Lane Merge Self-Attention, which computes geometric attention bias serving as an attention mask to enhance global information sharing. The resulting queries then perform cross-attention with BEV features. Then $Q_p$ and $Q_l$ together with $\hat{Q}_t$, are fed into Unified Scene Graph Network. The topology head computes point-lane topology, lane-lane topology and lane-traffic topology. During inference, predicted points and lanes are fused via Point-Lane Geometry Matching algorithm to refine lane endpoints and effectively mitigate the endpoint deviation problem.

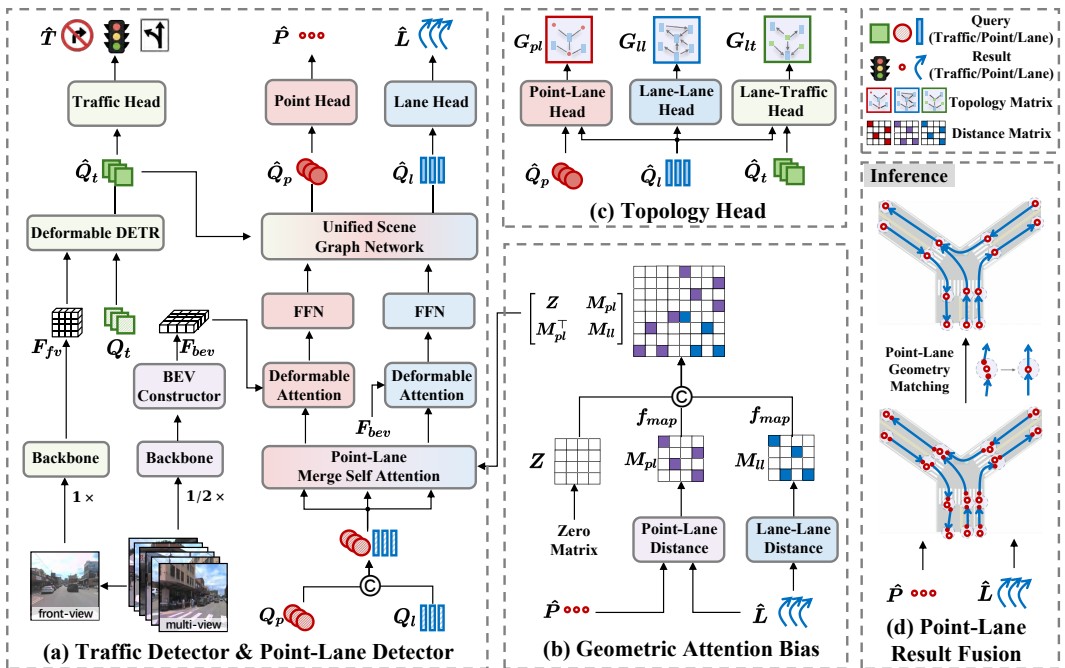

Figure 2: **TopoPoint framework.** (a) In addition to the traffic elements and lanes, lane endpoints are also explicitly perceived in the detector. (b) The geometric attention bias is also incorporated into the point-lane merge self attention module to exchange information. (c) On this basis, the queries are used for topology reasoning, and the topology is also used for query enhancement in scene graph network. (d) During inference, point-lane result fusion is applied to eliminate endpoint deviation.

## 3.3 Traffic Detector

To detect traffic elements in the front-view image, we initialize traffic element query $Q_t$, which interact with multi-scale front-view features $F_{fv}$ via Deformable DETR to compute cross-attention and produce updated representations $\hat{Q}_t$. The $\hat{Q}_t$ are then passed through the Traffic Head to predict 2D bounding boxes $\hat{T}$. The process is as follows:

$$\hat{Q}_t = \text{DeformableDETR}(Q_t, F_{fv}) \tag{1}$$

$$\hat{T} = \text{TrafficHead}(\hat{Q}_t) \tag{2}$$

where $Q_t \in \mathbb{R}^{N_t \times d}$, $F_{fv} \in \mathbb{R}^{H_F \times W_F \times d}$ and $\hat{T} \in \mathbb{R}^{N_t \times 4}$, $N_t$ denotes the number of $Q_t$, d denotes the feature dimension, $(H_{fv}, W_{fv})$ denotes the size of $F_{bev}$.

## 3.4 Point-Lane Detector

We independently initialize point query $Q_p$ and lane query $Q_l$. These queries first interact through Point-Lane Merge Self-Attention to exchange global information. The updated queries then compute cross-attention with the BEV features, followed by two separate feed-forward networks (FFNs). The resulting $Q_p$ and $Q_l$ are subsequently fed into Unified Scene Graph Network, where they aggregate features from each other via graph convolution networks (GCNs). The enhanced representations are finally used by the point head and lane head to regress endpoints and lane centerlines, respectively.

**Point-Lane Merge Self-Attention.** We first concatenate $Q_p$ and $Q_l$ along the instance dimension to form $Q_{pl}$. $Q_{pl}$ is then used as the query, key, and value in the self-attention computation. The definition of $Q_{pl}$ as follows:

$$Q_{pl} = \text{Concat}(Q_p, Q_l) \tag{3}$$

where $Q_p \in \mathbb{R}^{N_p \times d}$, $Q_l \in \mathbb{R}^{N_l \times d}$, $Q_{pl} \in \mathbb{R}^{N_{pl} \times d}$, $N_p$ denotes the number of $Q_p$, $N_l$ denotes the number of $Q_l$, $N_{pl} = N_p + N_l$ and $d$ denotes the feature dimension. To incorporate the geometric relationships between points and lanes in the BEV space, we compute their pairwise

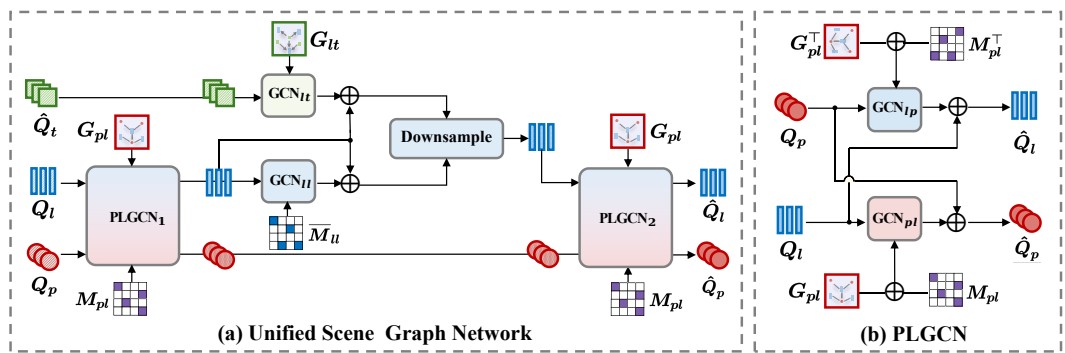

Figure 3: **Module details.** (a) Based on geometric attention bias and reasoned topology, lane & point queries are enhanced from the associated traffic elements & lanes & points by the unified scene graph network, (b) where the PLGCN is designed for better interaction between lanes and points.

geometric distances based on the predicted points $\hat{P}_{l-1} = \{\hat{p}_i \in \mathbb{R}^3 | i = 1, 2, \ldots, N_p\}$ and lanes $\hat{L}_{l-1} = \{\hat{l}_i \in \mathbb{R}^{k \times 3} | i = 1, 2, \ldots, N_l\}$ from the previous decoder layer, where k denote the number of points in each lane. These distances are then transformed by a learnable mapping function $f_{map}$ to obtain geometric bias matrix $M_{pp}$, $M_{pl}$ and $M_{ll}$, as follows:

$$D_{ll} = \left\{ \sum |\hat{l}_i^e - \hat{l}_j^s| \,\Big|\, i = 1, 2, \ldots, N_p, j = 1, 2, \ldots, N_l \right\} \tag{4}$$

$$D_{pl} = \left\{ \text{Min} \left( \sum |\hat{p}_i - \hat{l}_j^s|, \sum |\hat{p}_i - \hat{l}_j^e| \right) \,\Big|\, i = 1, 2, \ldots N_p, j = 1, 2, \ldots N_l \right\} \tag{5}$$

$$M_{pl} = f_{map}(D_{pl}), \; M_{ll} = f_{map}(D_{ll}) \tag{6}$$

where $\hat{l}_i^s \in \mathbb{R}^3$ denotes the start point of $\hat{l}_i$, $\hat{l}_i^e \in \mathbb{R}^3$ denotes the end point of $\hat{l}_i$, $D_{ll} \in \mathbb{R}^{N_l \times N_l}$ denote the L1 distance from the start points to the end points in $\hat{L}_{l-1}$, and $D_{pl} \in \mathbb{R}^{N_p \times N_l}$ denote the minimum L1 distance from $\hat{P}_{l-1}$ to the endpoints of $\hat{L}_{l-1}$. Notably, $f_{map} = e^{-\frac{x^P}{\lambda \cdot \hat{\sigma}}}$ is proposed in TopoLogic[15], $\alpha, \lambda$ are learnable parameters, and $\hat{\sigma}$ is the standard deviation of distance matrix $D$.

To compute self-attention, we concatenate $M_{pl}$, $M_{ll}$ to form geometric attention bias, which is added to the attention weights computed from $Q_{pl}$. The self attention process is described as follows:

$$Q_p, Q_l = \text{Softmax} \left( \frac{Q_{pl} \cdot Q_{pl}^\top}{\sqrt{d}} + \begin{bmatrix} Z & M_{pl} \\ M_{pl}^\top & M_{ll} \end{bmatrix} \right) \cdot Q_{pl} \tag{7}$$

$$Q_p, Q_l = \text{LN}(Q_p), \text{LN}(Q_p) \tag{8}$$

where $Z \in \mathbb{R}^{N_p \times N_p}$ denotes the zero matrix, $M_{pl} \in \mathbb{R}^{N_p \times N_l}$, $M_{ll} \in \mathbb{R}^{N_l \times N_l}$ and LN demotes the layer normalization.

**Point-Lane Deformable Cross Attention**. After self-attention, $Q_p$ and $Q_l$ are used to compute deformable cross-attention with the BEV feature. Specifically, we independently initialize two sets of learnable reference points, $R_p$ and $R_l$, corresponding to $Q_p$ and $Q_l$, which attends to the BEV feature via deformable cross-attention using its own reference points. The results are then passed through two separate feed-forward networks (FFNs). The process is described as follows:

$$Q_p, Q_l = \text{LN}(\text{DeformAttn}(Q_p, R_p, F_{bev})), \text{LN}(\text{DeformAttn}(Q_l, R_l, F_{bev})) \tag{9}$$

$$Q_p, Q_l = \text{LN}(\text{FFN}(Q_p)), \text{LN}(\text{FFN}(Q_l)) \tag{10}$$

where $R_p \in \mathbb{R}^{N_p \times 3}$, $R_l \in \mathbb{R}^{N_l \times 3}$, $F_{bev} \in \mathbb{R}^{H_B \times W_B \times d}$ denotes BEV feature map, $(H_B, W_B)$ denotes the BEV size of $F_{bev}$.

**Unified Scene Graph Network.** We construct a Unified Scene Graph Network by assembling the $Q_p$, $Q_l$, and $Q_t$, as illustrated in Figure 3(a). To enhance the interaction between point and lane representations, we further introduce the Point-Lane Graph Convolutional Network (PLGCN), as shown in Figure 3(b). The PLGCN is designed to facilitate bidirectional feature aggregation between $Q_p$ and $Q_l$ based on their geometric relationships. The structure of the PLGCN is as follows:

$$A_{pl} = \lambda_1 G_{pl} + \lambda_2 M_{pl} \tag{11}$$

$$Q_p = \text{GCN}_{pl}(Q_l, A_{pl}) + Q_p, \; Q_l = \text{GCN}_{lp}(Q_p, A_{pl}^\top) + Q_l \tag{12}$$

In the Unified Scene Graph Network, $Q_p$ and $Q_l$ first interact with each other through the first Point-Lane Graph Convolutional Network (PLGCN$_1$) to generate updated features $Q_p^1$ and $Q_l^1$. Then $Q_l^1$ is processed through two separate GCNs: GCN$_{ll}$ aggregates information from $Q_l^1$ itself to enhance intra-lane relationships, while GCN$_{lt}$ aggregates information from $\hat{Q}_t$ to incorporate semantic context. The outputs from these two branches are concatenated and downsampled to form $Q_l^2$. Finally, a second round of Point-Lane Graph Convolutional Network (PLGCN$_2$) is applied to $Q_l^2$ and $Q_p^1$, yielding the final enhanced features $Q_l^3$ and $Q_p^3$, which are used as the output of the Point-Lane detector decoder layer. The overall process can be formulated as:

$$Q_p^1, Q_l^1 = \text{PLGCN}_1(Q_p, Q_l, M_{pl}, G_{pl}) \tag{13}$$

$$Q_l^2 = \text{Downsample}\left(\text{Concat}\left(\text{GCN}_{ll}(Q_l^1, \overline{M}_{ll}) + Q_l^1, \; \text{GCN}_{lt}(\hat{Q}_t, G_{lt}) + Q_l^1\right)\right) \tag{14}$$

$$Q_p^3, Q_l^3 = \text{PLGCN}_2(Q_p^1, Q_l^2, M_{pl}, G_{pl}) \tag{15}$$

$$\hat{Q}_p, \hat{Q}_l = Q_p^3, Q_l^3 \tag{16}$$

where $\lambda_1, \lambda_2$ denotes the learnable parameters. $\text{GCN}(X, A) = \sigma(\hat{A}XW)$, $X$ denotes the input, $W$ denotes the learnable weight matrix, $A$ denotes the adjacency matrix, $\hat{A}$ denotes the normalized $A$ and $\sigma$ denotes sigmoid[42] function. $\overline{M}_{ll} = I + M_{ll} + M_{ll}^\top$, $I \in \mathbb{R}^{N_l \times N_l}$ denotes the identity matrix, $M_{pl}, M_{ll}$ is derived within the Point-Lane Merge Self-Attention, $G_{pl}, G_{lt}$ is derived within the Topology Head from the previous decoder layer. Downsample denotes the Linear-layer.

**Point-Lane Head.** After passing through the Unified Scene Graph Network, we obtain the enhanced point query $\hat{Q}_p$ and lane query $\hat{Q}_l$, which are fed into the PointHead and LaneHead, respectively, to produce the predicted point set $\hat{P} = \{\hat{P}_{reg}, \hat{P}_{cls}\}$ and lane set $\hat{L} = \{\hat{L}_{reg}, \hat{L}_{cls}\}$, as follows:

$$\hat{P} = \text{PointHead}(\hat{Q}_p), \; \hat{L} = \text{LaneHead}(\hat{Q}_l) \tag{17}$$

where $\hat{P}_{reg} \in \mathbb{R}^{N_p \times 3}$ and $\hat{L}_{reg} \in \mathbb{R}^{N_p \times k \times 3}$ denote the regressed points and lanes, respectively, $\hat{P}_{cls} \in \mathbb{R}^{N_p \times 1}$ and $\hat{L}_{cls} \in \mathbb{R}^{N_l \times 1}$ denotes classification scores for points and lanes, LaneHead and PointHead each consist of two separate MLP branches for regression and classification.

### 3.5 Topology Head

To predict the point-lane topology, lane-lane topology and lane-traffic topology. We perform topology reasoning based on the enhanced features $\hat{Q}_p$, $\hat{Q}_l$ and $\hat{Q}_t$ obtained from the detectors. We encode these features using separate MLPs and compute their pairwise similarities as the topology reasoning outputs. The process is formulated as follows:

$$\hat{G}_{pl} = \text{Sigmoid}(\text{MLP}(\hat{Q}_p) \cdot \text{MLP}(\hat{Q}_l)^\top) \tag{18}$$

$$\hat{G}_{ll} = \text{Sigmoid}(\text{MLP}(\hat{Q}_l) \cdot \text{MLP}(\hat{Q}_l)^\top) \tag{19}$$

$$\hat{G}_{lt} = \text{Sigmoid}(\text{MLP}(\hat{Q}_l) \cdot \text{MLP}(\hat{Q}_t)^\top) \tag{20}$$

where $\hat{G}_{pl} \in \mathbb{R}^{N_p \times N_l}$ denotes the point-lane topology, $\hat{G}_{ll} \in \mathbb{R}^{N_l \times N_l}$ denotes the lane-lane topology, $\hat{G}_{lt} \in \mathbb{R}^{N_l \times N_t}$ denotes the lane-traffic topology.

### 3.6 Training

During the training phase, the overall loss of TopoPoint is composed of detection loss and topology reasoning loss. The detection loss includes the traffic element detection loss, point detection loss and lane detection loss. The topology reasoning loss consists of the point-lane topology loss, lane-lane topology loss and lane-traffic topology loss. The total loss is defined as:

$$\mathcal{L}_{total} = \lambda_t \mathcal{L}_t + \lambda_p \mathcal{L}_p + \lambda_l \mathcal{L}_l + \lambda_{pl} \mathcal{L}_{pl} + \lambda_{ll} \mathcal{L}_{ll} + \lambda_{lt} \mathcal{L}_{lt} \tag{21}$$

where $\mathcal{L}_t$, $\mathcal{L}_p$ and $\mathcal{L}_l$ denote the traffic element detection loss, point detection loss and lane detection loss, respectively. $\mathcal{L}_{pl}$, $\mathcal{L}_{ll}$ and $\mathcal{L}_{lt}$ represent the losses for point-lane topology, lane-lane topology and lane-traffic topology reasoning. $\lambda_t$, $\lambda_p$, $\lambda_l$, $\lambda_{pl}$, $\lambda_{ll}$ and $\lambda_{lt}$ are the corresponding loss weights. Specially, the $\mathcal{L}_p$ and $\mathcal{L}_l$ consist of classification loss and regression loss, where the classification loss employs the Focal loss[43] and the regression loss utilizes the L1 loss[44]. For $\mathcal{L}_t$, in addition to classification loss and regression loss, we incorporate the GIoU loss[45] to further improve localization accuracy. For topology reasoning, we adopt the focal loss for both $\mathcal{L}_{pl}$, $\mathcal{L}_{ll}$ and $\mathcal{L}_{lt}$.

## 3.7 Inference

To mitigate the endpoint deviation issue in lane prediction during inference, we propose the Point-Lane Geometry Matching (PLGM) algorithm. This method first filters out high-confidence predictions from $\hat{P}_{reg}$ and $\hat{L}_{reg}$ using their associated classification scores $\hat{P}_{cls}$ and $\hat{L}_{cls}$. For each selected point $\hat{P}_i \in \hat{P}_{select}$, we identify a set of nearby lane endpoints $\mathcal{N}_i$ from $\hat{L}_{select}$ based on their geometric distances in the BEV space. If the matching is found, the selected point and its neighboring lane endpoints are jointly averaged to compute refined endpoint $\hat{E}_i$, which is then used to update the corresponding lane predictions. This refinement leads to better-aligned lane endpoints and improved overall topology consistency. The complete procedure is illustrated in Algorithm 1.

---

**Algorithm 1:** Point-Lane Geometry Matching Algorithm

---

**Input:** Predicted points $\hat{P}_{reg}, \hat{P}_{cls}$; predicted lanes $\hat{L}_{reg}, \hat{L}_{cls}$; classification thresholds $\tau_p, \tau_l$; geometry distance threshold $\delta$.

**Output:** Refined lanes $\hat{L}_{ref}$

**Step 1: High-Confidence Filtering**

Filter points with high classification scores: $\hat{P}_{select} = \{\hat{P}_{reg}^i \mid \hat{P}_{cls}^i > \tau_p\}$

Filter lanes with high classification scores: $\hat{L}_{select} = \{\hat{L}_{reg}^j \mid \hat{L}_{cls}^j > \tau_l\}$

**Step 2: Geometry-Based Matching and Refinement**

**foreach** *point* $\hat{P}_i \in \hat{P}_{select}$ **do**

    Initialize empty match set: $\mathcal{N}_i = \emptyset$ ;

    **foreach** *lane* $\hat{L}_j \in \hat{L}_{select}$ **do**

        **if** *distance*$(\hat{P}_i, \hat{L}_j^{endpoint}) < \delta$ **then**

            Add $\hat{L}_j$ to $\mathcal{N}_i$ ;

    **if** $\mathcal{N}_i \neq \emptyset$ **then**

        Compute refined endpoint:

        $\hat{E}_i = \frac{1}{|\mathcal{N}_i|+1} \left( \hat{P}_i + \sum_{\hat{L}_j \in \mathcal{N}_i} \hat{L}_j^{endpoint} \right)$;

        Update endpoints of all $\hat{L}_j \in \mathcal{N}_i$ with $\hat{E}_i$;

**return** $\hat{L}_{ref}$ *with refined endpoints*

---

where $\hat{P}_{reg} \in \mathbb{R}^{N_p \times 3}$, $\hat{L}_{reg} \in \mathbb{R}^{N_l \times k \times 3}$, $\hat{P}_{cls} \in \mathbb{R}^{N_p \times 1}$ and $\hat{L}_{cls} \in \mathbb{R}^{N_l \times 1}$. $N_p$ denotes the number of point query, $N_l$ denotes the number of lane query, and k denotes the number of points in each lane.

## 4 Experiment

### 4.1 Dataset and Metric

**Dataset.** We evaluate TopoPoint on the large-scale topology reasoning benchmark OpenLane-V2[17], which is constructed based on Argoverse2[46] and nuScenes[47]. The dataset provides comprehensive annotations for lane centerline detection, traffic element detection, and topology reasoning tasks. OpenLane-V2 is divided into two subsets: *subset_A* and *subset_B*, each containing 1,000 scenes captured at 2 Hz with multi-view images and corresponding annotations. Both subsets include annotations for lane centerlines, traffic elements, lane-lane topology, and lane-traffic topology. Notably, *subset_A* provides seven camera views as input, while *subset_B* includes six views.

**Metric.** We adopt the evaluation metrics defined by OpenLane-V2, including $DET_l$, $DET_t$, $TOP_{ll}$, and $TOP_{lt}$, all of which are computed based on mean Average Precision (mAP). Specifically, $DET_l$ quantifies similarity by averaging the Fréchet distance under matching thresholds of 1.0, 2.0, and 3.0. $DET_t$ evaluates detection quality for traffic elements using the Intersection over Union (IoU) metric, averaged across different traffic categories. $TOP_{ll}$ and $TOP_{lt}$ measure the similarity of the predicted lane-lane topology matrix and lane-traffic topology matrix, respectively. The overall OpenLane-V2 Score (OLS) is calculated as follows:

$$\text{OLS} = \frac{1}{4}[\text{DET}_l + \text{DET}_t + \sqrt{\text{TOP}_{ll}} + \sqrt{\text{TOP}_{lt}}] \tag{22}$$

All evaluation metrics are computed based on the latest version (v2.1.0) of OpenLane-V2, which is available on the official OpenLane-V2 GitHub repository. In addition, to evaluate the performance of endpoint detection, we define a custom metric $\text{DET}_p$, which is computed as the average over match thresholds $\mathbb{T} = \{1.0, 2.0, 3.0\}$ based on the point-wise Fréchet distance, as follows:

$$\text{DET}_p = \frac{1}{|\mathbb{T}|} \sum_{t \in \mathbb{T}} AP_t \tag{23}$$

### 4.2 Implementation Details

**Model details.** The multi-view images have a resolution of $2048 \times 1550$ pixels, with the front view specifically cropped and padded to match $2048 \times 1550$. Notably, all multi-view inputs are downsampled by a factor of 0.5 before being fed into the backbone, except for the front view, which is directly processed at the original resolution. A pretrained ResNet-50 is adopted as the backbone, and a Feature Pyramid Network is used as the neck to extract multi-scale features. The hidden feature dimension $d$ is set to 256. BEV grid size is configured to $200 \times 100$. The number of traffic element query $N_t$, point query $N_p$ and lane query $N_l$ are set to 100, 200 and 300, respectively. The sampled points number $k$ of each lane is set to 11. The decoder consists of 6 layers. Following TopoLogic, the learnable parameters $\lambda$ and $\alpha$ in the mapping function $f_{map}$ are initialized to 0.2 and 2.0, respectively, $\lambda_1$ and $\lambda_2$ in $A_{pl}$ are both initialized to 1.0. The detection loss weights $\lambda_t$, $\lambda_p$, $\lambda_l$ and are all set to 1.0, while the topology reasoning loss weights $\lambda_{ll}$ and $\lambda_{lt}$ are both set to 5.0. In inference, the classification thresholds for filtering high-confidence predictions are both set to $\tau_p = \tau_l = 0.3$. For geometric matching, the distance threshold $\delta$ is set to 1.5 meters to determine valid point-lane associations.

**Training details.** We train the traffic detector, point-lane detector and topology head in an end-to-end manner. TopoPoint is trained using the AdamW optimizer with a cosine annealing learning rate schedule, starting at $2.0 \times 10^{-4}$ with a weight decay of 0.01. All experiments are conducted for 24 epochs on 8 Tesla V100 GPUs with a batch size of 8.

### 4.3 Comparison on OpenLane-V2 Dataset

We compare TopoPoint with existing methods on the OpenLane-V2 benchmark, and the results are summarized in Table 1. On *subset_A*, TopoPoint achieves **48.8** on OLS, surpassing all previous approaches and achieving state-of-the-art performance. Notably, despite TopoFormer leveraging a pretrained lane detector, our method achieves superior performance (**48.8** v.s. 46.3 on OLS). Built upon TopoLogic, TopoPoint demonstrates superior performance in lane detection (**31.4** v.s. 29.9 on $\text{DET}_l$) and shows a substantial improvement in traffic element detection (**55.3** v.s. 47.2 on $\text{DET}_t$). Furthermore, it outperforms in lane-lane topology reasoning (**28.7** v.s. 23.9 on $\text{TOP}_{ll}$) and achieves better results in lane-traffic topology reasoning (**30.0** v.s. 25.4 on $\text{TOP}_{lt}$). Additionally, there is a notable improvement in the endpoint detection (**52.6** v.s. 45.2 on $\text{DET}_p$). Meanwhile, TopoPoint also achieves state-of-the-art performance on *subset_B* (**49.2** on OLS, **45.1** on $\text{DET}_p$), further demonstrating its effectiveness.

### 4.4 Ablation Study

We conduct ablation studies on several key components of TopoPoint using OpenLane-V2 *subset_A*.

**Impact of each module.** We conduct an ablation study to assess the impact of each module on topology reasoning performance. As shown in the Table 2, keeping the original front-view scale (scale =1.0) improves traffic element detection (**53.8** v.s. 46.8 on $\text{DET}_t$), enhancing lane-traffic topology reasoning (**27.0** v.s. 24.3 on $\text{TOP}_{lt}$). Adding Point-Lane Merge Self-Attention (PLMSA) boosts lane and endpoint detection (**30.2** v.s. 29.4 on $\text{DET}_l$, **49.8** v.s. 44.8 on $\text{DET}_p$), leading to better lane-lane and lane-traffic topology reasoning (**27.2** v.s. 23.8 on $\text{TOP}_{ll}$, **28.5** v.s. 27.0 on $\text{TOP}_{lt}$). Incorporating Point-Lane Graph Convolutional Network (PLGCN) further improves detection (**30.8** v.s. 30.2 on $\text{DET}_l$, **51.8** v.s. 49.8 on $\text{DET}_p$). Finally, the Point-Lane Geometry Matching (PLGM) algorithm refines lane endpoints during inference, mitigating endpoint deviation and enhancing lane and point detection (**31.4** v.s. 30.8 on $\text{DET}_l$, **52.6** v.s. 51.8 on $\text{DET}_p$).

**Effect of different GCNs.** We investigate the impact of various GCN designs on topology reasoning performance. As shown in Table 3, adding the lane-lane GCN and lane-traffic GCN improves lane

Table 1: Performance comparison on OpenLane-V2. Results are from TopoLogic and TopoFormer papers. TopoFormer* utilizes a pretrained lane detector. The $DET_p$ scores for TopoNet, TopoMLP, and TopoLogic are computed using their official codebases. "-" denotes the absence of relevant data.

| Data | Method | Conference | $DET_l\uparrow$ | $DET_t\uparrow$ | $TOP_{ll}\uparrow$ | $TOP_{lt}\uparrow$ | $OLS\uparrow$ | $DET_p\uparrow$ |
|------|--------|-----------|------|------|------|------|------|------|
| | STSU[13] | *ICCV2021* | 12.7 | 43.0 | 2.9 | 19.8 | 29.3 | - |
| | VectorMapNet[10] | *ICML2023* | 11.1 | 41.7 | 2.7 | 9.2 | 24.9 | - |
| | MapTR[48] | *ICLR2023* | 17.7 | 43.5 | 5.9 | 15.1 | 31.0 | - |
| *subset_A* | TopoNet[26] | *Arxiv2023* | 28.6 | 48.6 | 10.9 | 23.8 | 39.8 | 43.8 |
| | TopoMLP[29] | *ICLR2024* | 28.3 | 49.5 | 21.6 | 26.9 | 44.1 | 43.4 |
| | TopoLogic[15] | *NeurIPS2024* | 29.9 | 47.2 | 23.9 | 25.4 | 44.1 | 45.2 |
| | TopoFormer*[31] | *CVPR2025* | **34.7** | 48.2 | 24.1 | 29.5 | 46.3 | - |
| | TopoPoint (Ours) | - | 31.4 | **55.3** | **28.7** | **30.0** | **48.8** | **52.6** |
| | STSU[13] | *ICCV2021* | 8.2 | 43.9 | - | - | - | - |
| | VectorMapNet[10] | *ICML2023* | 3.5 | 49.1 | - | - | - | - |
| | MapTR[48] | *ICLR2023* | 15.2 | 54.0 | - | - | - | - |
| *subset_B* | TopoNet[26] | *Arxiv2023* | 24.3 | 55.0 | 6.7 | 16.7 | 36.8 | 38.5 |
| | TopoMLP[29] | *ICLR2024* | 26.6 | 58.3 | 21.0 | 19.8 | 43.8 | 39.6 |
| | TopoLogic[15] | *NeurIPS2024* | 25.9 | 54.7 | 21.6 | 17.9 | 42.3 | 39.2 |
| | TopoFormer*[31] | *CVPR2025* | **34.8** | 58.9 | 23.2 | 23.3 | 47.5 | - |
| | TopoPoint (Ours) | - | 31.2 | **60.2** | **28.3** | **27.1** | **49.2** | **45.1** |

Table 2: Ablation study on different modules. Baseline is reproduced using TopoLogic code.

| Module | $DET_l\uparrow$ | $DET_t\uparrow$ | $TOP_{ll}\uparrow$ | $TOP_{lt}\uparrow$ | $OLS\uparrow$ | $DET_p\uparrow$ |
|--------|------|------|------|------|------|------|
| Baseline | 29.2 | 46.8 | 23.4 | 24.3 | 43.4 | 44.5 |
| + FVScale | 29.4 | 53.8 | 23.8 | 27.0 | 46.0 | 44.8 |
| + PLMSA | 30.2 | 54.8 | 27.2 | 28.5 | 47.6 | 49.8 |
| + PLGCN | 30.8 | 55.3 | 28.0 | 29.2 | 48.3 | 51.8 |
| + PLGM | **31.4** | **55.3** | **28.7** | **30.0** | **48.8** | **52.6** |

Table 3: Ablation study on different GCNs. "w/o GCN" denotes removal of Unified Graph Network.

| Module | $DET_l\uparrow$ | $DET_t\uparrow$ | $TOP_{ll}\uparrow$ | $TOP_{lt}\uparrow$ | $OLS\uparrow$ | $DET_p\uparrow$ |
|--------|------|------|------|------|------|------|
| w/o GCN | 28.9 | 53.9 | 25.6 | 26.4 | 46.2 | 48.6 |
| + $GCN_{ll}$ | 29.8 | 54.2 | 26.9 | 27.1 | 47.0 | 49.8 |
| + $GCN_{lt}$ | 30.6 | 54.5 | 27.4 | 28.8 | 47.8 | 50.5 |
| + $PLGCN_1$ | 30.9 | 55.0 | 28.2 | 29.5 | 48.3 | 51.9 |
| + $PLGCN_2$ | **31.4** | **55.3** | **28.7** | **30.0** | **48.8** | **52.6** |

detection (**30.6** v.s. 29.8 v.s. 28.9 on $DET_l$), thereby enhancing both lane-lane and lane-traffic topology reasoning (**27.4** v.s. 26.9 v.s. 25.6 on $TOP_{ll}$, **28.8** v.s. 27.1 v.s. 26.4 on $TOP_{lt}$). Moreover, introducing two variants of the point-lane GCN effectively boosts both lane and endpoint detection performance (**31.4** v.s. 30.9 v.s. 30.6 on $DET_l$, **52.6** v.s. 51.9 v.s. 50.5 on $DET_p$).

**Image scales set up.** We investigate the impact of different image scaling strategies on topology reasoning performance. As shown in the Table 4, keeping the front-view image at its original resolution improves the performance of traffic element detection (**55.3** v.s. 48.6, **54.7** v.s. 48.3 on $DET_t$). On the other hand, downscaling the multi-view images by a factor of 0.5 slightly boosts lane detection performance (**31.2** v.s. 30.5, **31.4** v.s. 30.8 on $DET_l$).

**Effect of point and lane query numbers.** We investigate the impact of varying the number of point and lane query on topology reasoning performance. As shown in the Table 5, increasing the number of point query from 100 to 200 improves endpoint detection (**51.8** v.s. 49.7 on $DET_p$), which in turn enhances lane detection performance (**30.7** v.s. 29.5 on $DET_l$). However, further increasing the number from 200 to 300 introduces more negative point samples, leading to degraded endpoint detection (51.4 v.s. **52.6** on $DET_p$) and consequently worse lane detection performance (30.8 v.s. **31.4** on $DET_l$). On the other hand, increasing the number of lane query from 200 to 300 consistently improves lane detection accuracy(**31.4** v.s. 30.7 on $DET_l$).

## 4.5 Qualitative Results

Figure 4 provides a qualitative result comparison between TopoLogic and our TopoPoint. On the whole, both TopoLogic and TopoPoint yield good results. Nevertheless, as TopoLogic lacks a direct enhancement to lane detection itself, it is more likely to produce incorrect or missing lanes, thereby resulting in inaccurate or absent topologies. Benefit from the independent endpoint modeling and the

Table 4: Ablation study on front-view scale and multi-view scale. $S_{fv}$ denotes the scale of front-view, $S_{mv}$ denotes the scale of multi-view.

| $S_{fv}$ | $S_{mv}$ | $DET_l\uparrow$ | $DET_t\uparrow$ | $TOP_{ll}\uparrow$ | $TOP_{lt}\uparrow$ | OLS$\uparrow$ | $DET_p\uparrow$ |
|---|---|---|---|---|---|---|---|
| 0.5 | 0.5 | 31.2 | 48.6 | 28.5 | 28.4 | 46.6 | 52.3 |
| 0.5 | 1.0 | 30.5 | 48.3 | 28.0 | 27.9 | 46.1 | 51.5 |
| 1.0 | 0.5 | **31.4** | **55.3** | **28.7** | **30.0** | **48.8** | **52.6** |
| 1.0 | 1.0 | 30.8 | 54.7 | 28.3 | 28.9 | 48.1 | 51.8 |

Table 5: Ablation study on number of point query and lane query. $N_p$ denotes the number of point query, $N_l$ denotes the number of lane query.

| $N_p$ | $N_l$ | $DET_l\uparrow$ | $DET_t\uparrow$ | $TOP_{ll}\uparrow$ | $TOP_{lt}\uparrow$ | OLS$\uparrow$ | $DET_p\uparrow$ |
|---|---|---|---|---|---|---|---|
| 100 | 200 | 29.5 | 54.3 | 25.6 | 27.0 | 46.5 | 49.7 |
| 200 | 200 | 30.7 | 53.7 | 27.4 | 28.2 | 47.5 | 51.8 |
| 200 | 300 | **31.4** | **55.3** | **28.7** | **30.0** | **48.8** | **52.6** |
| 300 | 300 | 30.8 | 54.6 | 28.2 | 29.8 | 48.3 | 51.4 |

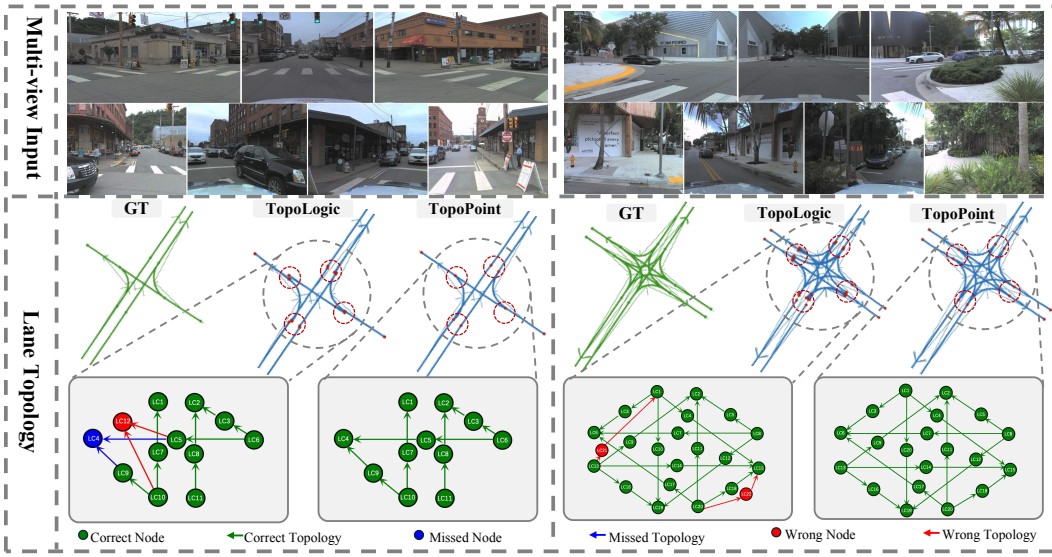

Figure 4: **Qualitative comparison of TopoLogic and our TopoPoint.** The first row denotes multi-view inputs, and the second row denotes lane detection result with lane topology result. In the graph form of lane topology, node indicates lane while edge indicates lane topology, where green/red/blue color respectively indicates the correct/wrong/missed prediction.

interaction between points and lanes, TopoPoint has managed to avoid such situations as much as possible. Moreover, it is evident that TopoPoint eradicates the endpoint deviation at lane connections, which still exist in TopoLogic. Both Figure 5 and Figure 6 provide more qualitative results comparison between TopoLogic and our TopoPoint.

## 5 Conclusion

In this paper, we identify the endpoint deviation issue in existing topology reasoning methods. To tackle this, we propose TopoPoint, which introduces explicit endpoint detection and strengthens point-lane interaction through Point-Lane Merge Self-Attention and Point-Lane GCN. We further design a geometry matching strategy to refine lane endpoints. Experiments on OpenLane-V2 show that TopoPoint achieves state-of-the-art performance in OLS. Additionally, we introduce $DET_p$ metric for evaluating endpoint detection, where TopoPoint also achieves significant improvement.

**Impact.** TopoPoint improves 3D lane detection by addressing endpoint deviation and enhancing topology reasoning, benefiting autonomous driving tasks like planning and mapping.

## 6 Acknowledgements

This work is supported by National Key R&D Program of China (2023YFD2000303) and National Natural Science Foundation of China (62372433).

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

Figure 5: **Additional qualitative comparison of TopoLogic and TopoPoint.** The first row denotes multi-view inputs, the second row denotes the endpoint detection and lane detection results, where the lane endpoints are indicated by red dots. The third row denotes the lane-lane topology result, and the last row denotes traffic element detection and lane-traffic topology results in the front-view.

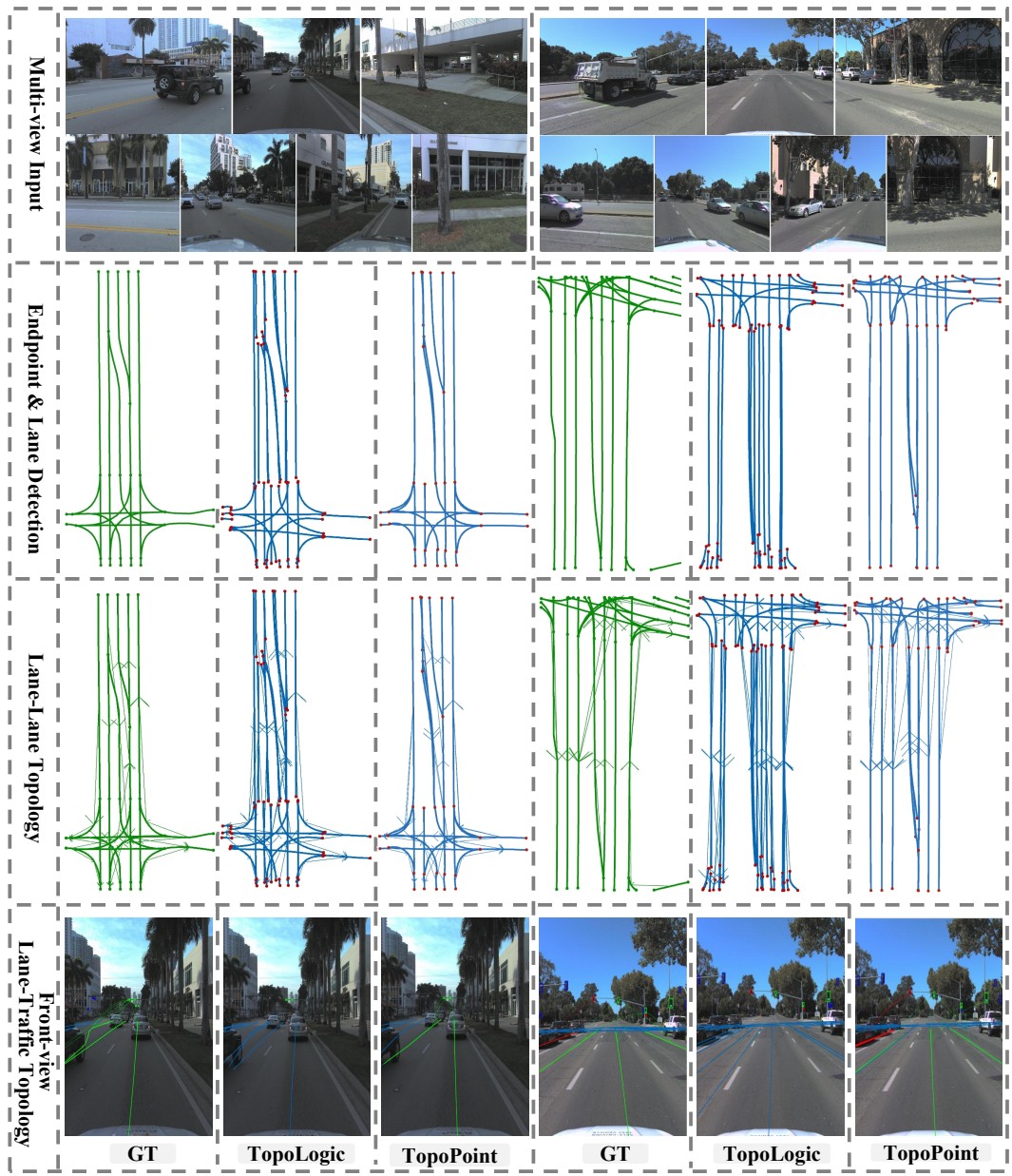

Figure 6: **More qualitative comparison of TopoLogic and TopoPoint.** The first row denotes multi-view inputs, the second row denotes the endpoint detection and lane detection results, where the lane endpoints are indicated by red dots. The third row denotes the lane-lane topology result, and the last row denotes traffic element detection and lane-traffic topology results in the front-view.

