# OpenReview forum: "TopoPoint: Enhance Topology Reasoning via Endpoint Detection in Autonomous Driving"
_NeurIPS.cc/2025/Conference — NeurIPS 2025 poster_

### Official Review · Reviewer_HgsV · 2025-06-02

**Clarity:** 4
**Significance:** 3
**Originality:** 3
**Rating:** 5
**Confidence:** 3

**Summary:**

This paper aims to solve the endpoint deviation problem in lane-phase inference, which is central to understanding autonomous intersections.  The authors added a separate branch for detecting endpoints and designed Point-Lane Merge Self-Attention and Point-Lane  Graph Convolutional Network to exchange global information between endpoints and lane queries.  During the inference stage, the predicted lane endpoints are aligned with the detected endpoints through geometric matching to correct remaining errors. The method demonstrated its effectiveness by achieving state-of-the-art performance on the OpenLane-V2 benchmark.  The introduction of the $DET_p$ metric further enhances the work's originality by providing a new tool for evaluating a previously less quantified aspect of lane detection performance.

**Questions:**

1. Analysis of Inter-Module Error Propagation:
Whether the performance degradation of ($DET_t$) only affects the lane-traffic topology ($TOP_lt$), or whether it can also negatively affect the geometric accuracy of lane($DET_l$) and endpoint ($DET_p$) predictions through feature propagation.

2. Verification of Generalization Performance on Other Datasets:
All experiments in this paper were performed only on the OpenLane-V2 benchmark. To prove the generalizability of the TopoPoint methodology, we would like to know if there are additional experimental results on other tasks in nuScenes or Argoverse2, or on different datasets (e.g., Waymo Open Dataset).

3. Sensitivity on calibration accuracy:
TopoPoint explicitly acknowledges that it relies on accurate calibration, could you provide sensitivity analysis experimental results showing the changes in each evaluation indicator for each correction error range.

**Ethical Concerns:**

["NO or VERY MINOR ethics concerns only"]

**Final Justification:**

It would be good to add the above-mentioned experiments to the paper. The rebuttal addressed my concerns. I am keeping my rating as positive.

**Limitations:**

yes

**Paper Formatting Concerns:**

1. Tabel 1: Is there any reason for mentioning the conference in Table 1

**Quality:**

3

**Strengths And Weaknesses:**

[Strengths]
1. Novel Solution for endpoint detection in autonomous driving:
This paper identifies and emphasizes the importance of the “endpoint deviation” problem, which is practical but often overlooked in the field of autonomous driving topology inference. It specifically explains how minor discrepancies at the endpoints that occur when predicting multiple lanes independently can cause problems for subsequent modules. As a solution, the core idea of explicitly detecting lane endpoints as separate, independent entities and using them to refine lane predictions and enforce topological consistency is highly innovative and directly addresses the identified problem.
The core idea of explicitly detecting lane endpoints as separate entities and then using them to refine lane predictions and reason about topology is novel and directly tackles this deviation problem.

2. Novel metric for endpoint detection:
To directly demonstrate the improvement in endpoint detection accuracy, we propose a new evaluation metric, $DET_p$. This metric shows a significant performance improvement over existing methods, providing strong quantitative evidence to support the claims made in this paper, from problem definition to verification.

3. Detailed and Reproducible Methodology:
The paper provides detailed explanations and mathematical formulas for each component of the proposed TopoPoint framework. From the geometric bias calculation of PLMSA to the complex GCN interactions within USGN, the learning loss function configuration, and hyperparameter settings, it provides sufficient information necessary for research reproduction.

[Weaknesses]
1. Lack of Real-time and Computational Cost Analysis:
There is no analysis of the inference speed (FPS) or computational cost (GFLOPs) of the proposed model. The TopoPoint architecture, especially USGN, which includes multiple GCN modules, is conceptually complex and is expected to require significant computational power. Real-time performance is critical for autonomous driving systems, so it is necessary to analyze whether the performance improvement compared to simpler models justifies the additional computational cost.

2. Proof of Impact on Downstream Tasks:
The paper argues that solving the endpoint deviation problem is important for subsequent path planning and control tasks, but it does not provide direct experimental evidence to support this claim. If a quantitative evaluation of how much improved topological accuracy actually improves the performance of the planning module had been included, the practical usefulness of the proposed method and the logical persuasiveness of the paper would have been greatly enhanced.

3.  Lack of calibration sensitivity analysis:
In their conclusion, the authors acknowledged that TopoPoint relies on accurate camera calibration, but they did not provide a sensitivity analysis of how performance degrades depending on the degree of calibration error. This is important information for evaluating the robustness of the model to sensor errors that may occur in actual vehicle applications.

---

> ### Author Rebuttal · Authors · 2025-07-30
>
> **Thank you very much for your high recognition of our work. We hope our response and clarifiction can ease some of your concerns.**
>
> ----
>
> **Q1:**  Lack of Real-time and Computational Cost Analysis: The TopoPoint architecture, especially USGN, which includes multiple GCN modules, is conceptually complex and is expected to require significant computational power. It is necessary to analyze whether the performance improvement compared to simpler models justifies the additional computational cost.
>
> **A1**: As you noted, real-time performance is crucial for application deployment. Following your suggestion, we evaluated the model's computational resources on Tesla A100 GPU, with the results as follows:
>
> | Model                | FLOPs  | #PARAMS | FPS  |
> | -------------------- | ------ | ------- | ---- |
> | TopoNet              | 712.1G | 38.6M   | 16.6 |
> | TopoLogic            | 665.0G | 37.8M   | 17.1 |
> | TopoPoint (w/o USGN) | 973.8G | 66.3M   | 15.3 |
> | TopoPoint            | 989.3G | 67.9M   | 14.8 |
>
> - **As shown in the table, GCN does not actually consume significant computational resources, and the introduction of USGN has a negligible impact on FLOPs, #PARAMS, and FPS.**
> - This is because GCN is merely a method for aggregating node information, whose operations consist of simple matrix multiplications with any complex operation, i.e.,  $GCN(X, A) = \sigma(\hat{A}XW )$ as described in Line 168-170 in the paper. Furthermore, in the topology reasoning task, the number of nodes involved in GCN is equal to the number of queries ($N_t=100, N_p=200, N_l=300$), which actually has very low computational cost and small number of parameters..
>
> ----
>
> **Q2:** Proof of Impact on Downstream Tasks:  If a quantitative evaluation of how much improved topological accuracy actually improves the performance of the planning module had been included, the practical usefulness of the proposed method and the logical persuasiveness of the paper would have been greatly enhanced.
>
> **A2**: Previous methods often suffer from significant endpoint deviations in lane predictions, resulting in non-smooth connections between multi-segment lanes. Directly using such predictions for vehicle planning and control leads to frequent and unnecessary steering maneuvers, necessitating additional post-processing to mitigate these issues. In contrast, our method effectively eliminates endpoint deviations, producing smoothly connected lanes—as evidenced by the visualizations in Figure 4 and the supplementary materials—thereby providing more reliable inputs for downstream planning and control tasks. **Furthermore, experimental results in Table 1 of paper "*Reasoning multi-agent behavioral topology for interactive autonomous driving*" demonstrate that improved topological reasoning directly contributes to better performance in downstream planning tasks.**
>
> [1] H. Liu, L. Chen, Y. Qiao, C. Lv, and H. Li, “Reasoning multi-agent behavioral topology for interactive autonomous driving,” in Annual Conference on Neural Information Processing Systems, 2024.
>
> -----
>
> **Q3:** Analysis of Inter-Module Error Propagation: Whether the performance degradation of $DET_t$ only affects the lane-traffic topology $TOP_{lt}$, or whether it can also negatively affect the geometric accuracy of lane $DET_l$ and endpoint $DET_p$ predictions through feature propagation.
>
> **A3:** This is an interesting question, and our answer is almost no.
>
> - From a logical perspective: Traffic and Lane are mutually independent entities, and only the $TOP_{lt}$ metric evaluates the topological quality between them. Thus, the performance of Traffic only affects $DET_t$ and $TOP_{lt}$, but not $DET_l$ or $TOP_{ll}$.
> - From the model methodology: **Traffic and Lane are predicted through separate branches from independent queries, with no interaction at the feature level and no cross-branch influence. The only interaction is that GCN$_{lt}$ in USGN enhances Lane queries using Traffic**. Additionally, end-to-end joint optimization will eventually result in slight influences between components.
> - From the experimental results: According to Table 2, after adding FVScale, $DET_t$ and $TOP_{lt}$ show significant improvements (46.8→53.8, 24.3→27.0), while $DET_l$, $TOP_{ll}$, and $ DET_p$ are slightly affected (29.2→29.4, 23.4→23.8, 44.5→44.8), which verifies their weak correlation of influence. According to Table 3, after adding $GCN_{lt}$, the most affected is $TOP_{lt}$ (27.1→28.8), while the impact on other components is minor (29.8→30.6 on $DET_l$, 54.2→54.5 on $DET_t$, 26.9→27.4 on $TOP_{ll}$).
>
> ----
>
> **Q4:** Verification of Generalization Performance on Other Datasets: All experiments in this paper were performed only on the OpenLane-V2 benchmark. To prove the generalizability of the TopoPoint methodology, we would like to know if there are additional experimental results on other tasks in nuScenes or Argoverse2, or on different datasets (e.g., Waymo Open Dataset).
>
> **A4**: Adequate experiments are crucial, and this question can be answer from three aspects:
>
> - **First, OpenLane-v2 actually consists of two independent data subsets: subset_A is built upon Argoverse2, while subset_B is built upon nuScenes. To some extent, already demonstrates its generalization ability.**
>
> - Like other works such as TopoNet and TopoLogic, our research focuses on the topology reasoning task and is not applicable to other tasks in nuScenes or Argoverse2, so we cannot provide evaluations on those other tasks.
>
> - We also hope to have more data for research. Unfortunately, due to the constraints of annotation costs, there are currently no more datasets in the emerging field of topology reasoning. OpenLane-v2 is already the large and widely recognized dataset in this area, and all topology reasoning methods are evaluated on this benchmark. Although the Waymo Open Dataset is also an autonomous driving dataset, it don't providing topology annotations, which makes evaluations impossible.
>
> ----
>
> **Q5:** Sensitivity on calibration accuracy: TopoPoint explicitly acknowledges that it relies on accurate calibration, could you provide sensitivity analysis experimental results showing the changes in each evaluation indicator for each correction error range.  This is important information for evaluating the robustness of the model to sensor errors that may occur in actual vehicle applications.
>
> **A5**: As mentioned in our paper, accurate camera calibration is also important for the model. Camera parameters affect the accuracy of BEV features, which in turn impacts the performance of lane detection and endpoint detection. **To evaluate this effect, we introduced noise to the extrinsic parameters of the camera, adding translation and rotation noise with standard deviations of (10 cm, 3°) and (20 cm, 5°)**, respectively, with the experiment results as follows:
>
> | Model | $DET_l$ | $DET_t$ | $TOP_{ll}$ | $TOP_{lt}$ | $OLS$ | $DET_p$ |
> | --------- | -------- | -------- | -------- | -------- | -------- | -------- |
> | TopoPoint | **31.4** | **55.3** | **28.7** | **30.0** | **48.8** | **52.6** |
> | 10 cm, 3° | 30.3 | 55.1 | 27.2 | 28.8 | 47.8 | 50.8 |
> | 20 cm, 5° | 28.6 | 54.8 | 25.5 | 27.0 | 46.5 | 48.7 |
>
> - **It can be observed that as the noise increases, the performance of both lane and endpoint detection degrades accordingly, which further affects the overall topology reasoning. In contrast, traffic element detection remains largely unaffected, as it is performed directly in the front-view image space and does not rely on BEV features.**
>
> - This is a common challenge faced by topology reasoning and even 3D perception, and is not limited to our method.
>
> - While it shows some sensitivity to calibration accuracy, this issue can be effectively addressed via accurate online camera calibration during deployment.
>
> **Q6:**  Is there any reason for mentioning the conference in Table 1.
>
> **A6**:  We included the conference venues in Table 1 to **highlight that the compared methods are representative and state-of-the-art approaches recognized by the research community.** This underscores the rigor of our experimental setup and the competitiveness of our proposed method.
>
> **If you have any further questions or suggestions about our article, we'd love to discuss our content with you.**

---

> ### Comment · Reviewer_HgsV · 2025-08-07
>
> It would be good to add the above-mentioned experiments to the paper. The rebuttal addressed my concerns. I am keeping my rating as positive.

---

> > ### Author Response · Authors · 2025-08-08
> >
> > Thank you for your valuable suggestion. We will include the mentioned experiments in the final version of the paper. We sincerely appreciate your insightful comments and the issues you raised regarding our work.

---

### Official Review · Reviewer_P95Y · 2025-07-01

**Clarity:** 3
**Significance:** 2
**Originality:** 2
**Rating:** 4
**Confidence:** 2

**Summary:**

This paper introduces TopoPoint, a framework aimed at improving topology reasoning in autonomous driving by explicitly detecting lane endpoints and jointly modeling lanes and endpoints. The authors identify the persistent issue of endpoint deviation in existing lane topology models and address it by proposing: (1) Point-Lane Merge Self-Attention for to exchange global information between endpoints and lanes, (2) Point-Lane Graph Convolutional Network for message passing between lanes and endpoints, and (3) Point-Lane Geometry Matching for refinement during inference. Evaluations on the OpenLane-V2 dataset demonstrate state-of-the-art performance across multiple metrics, including the newly proposed $DET_p$ metric for endpoint detection. Ablation studies show consistent improvements from each module.

**Questions:**

* Given the complexity of TopoPoint, can the authors provide runtime benchmarks? Is the system deployable at real-time rates?
* Have the authors considered evaluating TopoPoint on datasets with different geographic or environmental conditions (e.g., KITTI, DENSE)? Qualitative or quantitative comparisons on such datasets would provide insights on the robustness of TopoPoint.

**Ethical Concerns:**

["NO or VERY MINOR ethics concerns only"]

**Final Justification:**

The paper introduces novel contributions with strong results and ablations. While I understand that single-run experiments are common in prior works, I believe this is not the best scientific practice. Therefore, I'm keeping my current (overall positive) assessment.

**Limitations:**

yes

**Quality:**

3

**Strengths And Weaknesses:**

**Strengths**
* Results on OpenLane-V2 are consistently strong, outperforming prior works such as TopoLogic and TopoFormer across the sub-tasks.
* The improvements brought by each component of TopoPoint are well-documented in the ablation study.

**Weaknesses**
* The use of a fixed number of queries could limit adaptability in dense or highly dynamic traffic scenes, as noted in the paper.
* As acknowledged in the checklist, all results are based on single experiment runs without variance estimates.
* The method is evaluated exclusively on data from Argoverse and nuScenes (U.S. and Singapore) under primarily favorable weather. This leaves open questions about performance under adverse conditions or in new geographical domains.

---

> ### Author Rebuttal · Authors · 2025-07-30
>
> **Thanks for your careful reading of our paper. We hope our response and clarifiction can ease some of your concerns.**
>
> **Q1:** Given the complexity of TopoPoint, can the authors provide runtime benchmarks? Is the system deployable at real-time rates?
>
> **A1:** As you noted, real-time performance is an important consideration for deployment. Following your suggestion, we evaluated the computational cost of our model on Tesla A100 GPU. **The results show that although TopoPoint introduces slightly higher resource consumption compared to the baseline, the increase is marginal.** Thanks to its compact parameter size, TopoPoint remains easy to deploy and demonstrates favorable real-time performance.
>
> | Model     | FLOPs  | #PARAMS | FPS  |
> | --------- | ------ | ------- | ---- |
> | TopoNet   | 712.1G | 38.6M   | 16.6 |
> | TopoLogic | 665.0G | 37.8M   | 17.1 |
> | TopoPoint | 989.3G | 67.9M   | 14.8 |
>
> ----
>
> **Q2:** Have the authors considered evaluating TopoPoint on datasets with different geographic or environmental conditions (e.g., KITTI, DENSE)? Qualitative or quantitative comparisons on such datasets would provide insights on the robustness of TopoPoint. This leaves open questions about performance under adverse conditions or in new geographical domains.
>
> **A2**:In terms of practical applications, adapting to different weather conditions and regions is crucial. Unfortunately, due to the constraints of annotation costs, there are no additional datasets available in the emerging field of topology reasoning at present. OpenLane-v2 is already the largest and widely recognized dataset to date. **While datasets like KITTY and DENSE are also related to autonomous driving, they do not provide topological annotations, which makes it impossible to conduct evaluations for the time being.**  Moreover, the camera configurations in the KITTI and DENSE datasets differ significantly from those in OpenLane-V2, making it infeasible to directly apply TopoPoint for inference on these datasets.
>
> In the future, we will also take these factors into account to promote the practical implementation of topology reasoning.
>
> ----
>
> **Q3:** The limitation in Limimataion Section or checklist:
>
> - The use of a fixed number of queries could limit adaptability in dense or highly dynamic traffic scenes.
> - All results are based on single experiment runs without variance estimates.
>
> **A3**: These limitations are indeed mentioned in the paper, but they are prevalent across various research efforts in the field of topology reasoning.
>
> - Currently, mainstream topology reasoning models are all based on the DETR-like architecture, where the number of queries can only be fixed at present.
>
> - **Existing topology reasoning methods (e.g. TopoNet, TopoMLP, TopoLogic) are evaluated based on single experiments. For one thing, single experiments on large-scale datasets can basicly verify the effectiveness;** for another, multiple experiments with deep learning methods consume significant computational resources.
>
> **We hope this response could help address your concerns, and wish to receive your further feedback soon.**

---

> > ### Comment · Reviewer_P95Y · 2025-08-02
> >
> > I thank the authors for their response. I maintain my positive assessment of the submission.

---

> > > ### Author Response · Authors · 2025-08-06
> > >
> > > Thank you very much for taking the time to carefully read our response. We are honored to hear your positive assessment of our response. If you have any further questions, we are more than happy to discuss with you if time permits. If you are satisfied with our response, we kindly hope that you would consider increasing our score.

---

### Official Review · Reviewer_gBnP · 2025-07-02

**Clarity:** 2
**Significance:** 3
**Originality:** 3
**Rating:** 4
**Confidence:** 3

**Summary:**

This paper focuses on the topology reasoning task in autonomous driving. Previous methods first predict independent lanes and then use a topology head to generate a lane-lane adjacency matrix, but they suffer from endpoint deviation (or inconsistent endpoints). To address this, the authors propose jointly predicting lanes and endpoints, then matching lanes to shared endpoints to ensure consistency. Experiments on the OpenLane-V2 benchmark demonstrate that the proposed method outperforms SOTA methods.

**Questions:**

1. Conduct more fair comparison by removing the image scale trick or adding this trick to SOTA methods.
2. The authors should analyze why TopoPoint underperforms TopoFormer in DET_l (lane detection accuracy). Can TopoFormer's architecture benefit from the proposed endpoint strategy to achieve comparable gains?
3. What about adopt PLGM to baseline methods? If lane detection is sufficiently accurate, the Point-Lane Geometry Matching (PLGM) module could be independently validated on baseline methods (without endpoint detection) to assess its standalone contribution to topology reasoning.

**Ethical Concerns:**

["NO or VERY MINOR ethics concerns only"]

**Final Justification:**

This paper proposes a effective method to improve the topology reasoning performances, and the author's rebuttal addressed my concerns. I keep my positive rating.

**Limitations:**

Yes.

**Paper Formatting Concerns:**

no formatting issues.

**Quality:**

3

**Strengths And Weaknesses:**

Strengths
1. The idea of introducing endpoints to improve lane topology reasoning is both intuitive and effective.
2. The proposed method achieves state-of-the-art performance on the OpenLane-V2 benchmark.
3. Ablation studies demonstrate that the proposed components (point-lane attention, point-lane GCN, and geometry matching) each contribute to improving OLS performance.

Weaknesses
1. The authors use high-resolution front-view images while downsampling other views by 0.5×, which alone improves OLS from 43.4 to 46.0. To isolate the impact of endpoint detection, this strategy should either be removed or applied uniformly to SOTA methods. Otherwise, TopoPoint's gains may be attributed to this resolution trick rather than topology reasoning.
2. Despite superior topology metrics, TopoPoint lags behind TopoFormer in DET_l. The author should explain why the proposed method can no achieve better lane detection performances.

---

> ### Author Rebuttal · Authors · 2025-07-30
>
> **Thank you very much for carefully reviewing our paper and providing in-depth feedback. We hope our response and clarifiction can ease some of your concerns.**
>
> **Q1:** TopoPoint's gains may be attributed to this resolution rather than topology reasoning. More fair comparison by removing the image scale or adding this trick to SOTA methods need to be conducted.
>
> **A1:** We fully understand your concerns about the FV resolution. Fortunately, we have conducted comprehensive ablation studies on the paper (as shown in Tables 2 & 4 of the paper).
>
> - First, it is important to clarify that OLS is an aggregated metric (calculated as $OLS = DET_l + DET_t + \sqrt{TOP_{ll}} + \sqrt{TOP_{lt}}$). Its improvement can be traced back to other sub-metrics, allowing us to understand the source of the improvement.
>
> - According to the results in Table 4, using a front view with 1.0x resolution mainly affects traffic-related metrics (48.6 → 55.3 on $DET_t$, 28.4 → 30.0 on $TOP_{lt}$), while having little impact on lane-related metrics (31.2 → 31.4 on $DET_l$, 28.5 → 28.7 on $TOP_{ll}$). Additionally, similar results are observed in Table 2 between the *Baseline* and *Baseline+FVScale*. **The core of TopoPoint lies in the explicit modeling and co-optimization of lane endpoints, which is primarily reflected in the metrics of $DET_l$, $TOP_{ll}$, and $TOP_{lt}$ . This improvement can be considered independent and orthogonal to that brought by resolution adjustments.**
>
> - The results in the first row of Table 4 ($S_{fv}=0.5$, $S_{mv}=0.5$) can actually be fairly compared with other methods in Table 1,  and some results are  integrated from Tables 1 and 4 as follows:
>
>   | Model | $DET_l$ | $DET_t$ | $TOP_{ll}$ | $TOP_{lt}$ | $OLS$ | $DET_p$ |
>   | ---------- | ----------- | ------------ | ---- | ---------- | ----------- | ---------- |
>   | TopoLogic (Tabel1)                    | 29.9  | 47.2  | 23.9   | 25.4   | 44.1 | 45.2 |
>   | TopoPoint (Table4, FV-0.5x)              | 31.2  | 48.6  | 28.5   | 28.4   | 46.6 | 52.3 |
>   | TopoPoint (Tabel4, FV-1.0x) | **31.4** | **55.3** | **28.7** | **30.0** | **48.8** | **52.6** |
>
> - **Increasing the front view resolution is simple to implement, yields significant effects, and incurs little computational overhead since it is only used in the traffic element branch, as verified by follow table.** Regrettably, this noteworthy strategy has been overlooked by previous studies. Therefore, we hope to share this strategy to promote research and applications in this field.
> | Model     | FLOPs  | #PARAMS | FPS  |
> | --------- | ------ | ------- | ---- |
> | TopoPoint (FV-0.5x) | 989.3G | 67.9M   | 14.8 |
> | TopoPoint (FV-1.0x) | 1077.7G | 67.9M   | 14.4 |
>
>
> ----
>
> **Q2:** Why TopoPoint underperforms TopoFormer in $DET_l $ (lane detection accuracy)? Can an TopoFormer's architecture benefit from the proposed endpoint strategy to achieve comparable gains?
>
> **A2**: To clarify why TopoFormer achieves a higher $DET_l$, it is necessary to elaborate on its training strategy, which is distinctly different from that of other methods.
>
> - **As noted in Section 4.3 (lines 243–244) and the caption of Table 1, TopoFormer employs a two-stage training strategy involving a pre-trained lane detector. In contrast, other baselines as well as our proposed  TopoPoint are trained using a simpler end-to-end approach, as described in Section 4.2 (line 236)**. This fundamental difference in training strategy gives TopoFormer an inherent advantage in lane-related metrics, and should be taken into account when comparing performance.
> - Theoretically, TopoFormer can benefit from the proposed endpoint strategy. Unfortunately, perhaps because CVPR 2025  was held in June, the official repository provided by TopoFormer has not included any code as of today (only a README), and the training details of the pre-trained detector have not been disclosed. Therefore, we cannot conduct effective verification on this for the time being.
>
> ----
>
> **Q3:** What about adopt PLGM to baseline methods?
>
> **A3**: This idea is very constructive, as it means that PLGM can be directly integrated into other models as a training-free component to improve performance. We conducted experiments as shown in the table below.
>
>  | Model | $DET_l$ | $DET_t$ | $TOP_{ll}$ | $TOP_{lt}$ | $OLS$ | $DET_p$ |
>  | ----------------- | -------- | -------- | -------- | -------- | -------- | -------- |
>  | TopoNet           | 28.6     | 48.6     | 10.9     | 23.8     | 39.8     | 43.8     |
>  | TopoNet (+PLGM)   | 29.3     | 48.6     | 11.7     | 24.6     | 40.4     | 44.9     |
>  | TopoLogic         | 29.9     | 47.2     | 23.9     | 25.4     | 44.1     | 45.2     |
>  | TopoLogic (+PLGM) | 30.5     | 47.2     | 24.6     | 26.0     | 44.6     | 46.1     |
>  | TopoPoint         | **31.4** | **55.3** | **28.7** | **30.0** | **48.8** | **52.6** |
>
> The results indicate that the introduction of PLGM can yield improvements in TopoNet and TopoLogic. Further improvements still require explicitly modeling and learning endpoints, thereby enhancing performance within the model itself.
>
> **We hope this response could help address your concerns, and wish to receive your further feedback soon.**

---

> > ### Comment · Reviewer_gBnP · 2025-08-04
> >
> > The rebuttal addressed my concerns. I keep my rating as positive.

---

> > > ### Author Response · Authors · 2025-08-06
> > >
> > > Thank you for carefully reviewing our response. We are honored to hear that our reply has addressed your concerns. If you have any further questions, we would be more than happy to discuss with you. If you are satisfied with our response, we kindly hope you would consider increasing our score.

---

### Official Review · Reviewer_VkTs · 2025-07-03

**Clarity:** 3
**Significance:** 3
**Originality:** 3
**Rating:** 4
**Confidence:** 5

**Summary:**

This paper proposes a topology reasoning method in Bird’s-Eye View (BEV). The core idea is to address the problem of *diverged endpoints*—where lanes that should share the same endpoints are predicted as having different endpoints. The authors introduce corresponding network designs, including point query, point-lane decoder, and geometry matching, which are technically sound.

**Questions:**

Although this method achieves competitive performance compared to existing approaches, the accuracy of HD topology reasoning remains far from practical application. I would like to hear the authors’ thoughts on the following considerations:

1. **Data Sufficiency Hypothesis**: Are current methods theoretically sufficient for HD topology reasoning, with performance limitations primarily due to insufficient training data? Would more data alone resolve these issues?
2. **Inherent Limitations Hypothesis**: Even with abundant data, might current methods still fail to fully address the problem, suggesting fundamental limitations in the approach?

**Ethical Concerns:**

["NO or VERY MINOR ethics concerns only"]

**Final Justification:**

The overall method is technically sound and good. I have no further questions and I keep my positive rating.

**Limitations:**

yes

**Quality:**

3

**Strengths And Weaknesses:**

The quality of this work is high. The paper is well-written and easy to follow. The proposed design effectively tackles topology reasoning challenges and holds significant value. The originality of the approach is relatively strong.

---

> ### Author Rebuttal · Authors · 2025-07-30
>
> **Thank you very much for high recognition  on our work. We are glad to discuss these open questions you raised with you.**
>
> **Q1:** Although this method achieves competitive performance compared to existing approaches, the accuracy of HD topology reasoning remains far from practical application. I would like to hear the authors’ thoughts on the following considerations:
>
> 1. **Data Sufficiency Hypothesis**: Are current methods theoretically sufficient for HD topology reasoning, with performance limitations primarily due to insufficient training data? Would more data alone resolve these issues?
> 2. **Inherent Limitations Hypothesis**: Even with abundant data, might current methods still fail to fully address the problem, suggesting fundamental limitations in the approach?
>
> **A1:** We sincerely thank you for the insightful question. Regarding the two hypotheses you raised, we believe that the performance of HD topology reasoning is currently constrained by both the scale of available data and limitations of existing model architectures. We address each point below:
>
> - **On the role of data**: Increasing the amount of training data does lead to improved performance. For a complex task like HD topology reasoning, large-scale and diverse datasets are essential for achieving robust generalization. However, it is equally important to ensure consistent and high-quality annotations, such as precise definitions of lane endpoints and other topology-critical elements, during data collection and labeling.
> - **On model limitations**: Current methods still have considerable room for improvement. Many existing approaches rely on DETR-like architectures to detect lane centerlines, which may not be fully reliable. Similarly, using simple MLPs to model topological structures can fall short in capturing the complex spatial and relational dependencies inherent in driving scenes. As research progresses, we anticipate that more specialized and expressive architectures will emerge to better capture the nuances of topology reasoning.
> - **On system-level iteration**: Building a reliable AI system requires co-evolution of both data and model design. Our proposed method has demonstrated its effectiveness on a reasonably large-scale benchmark (OpenLane-V2), achieving state-of-the-art performance. However, we acknowledge that broader deployment in industrial-scale settings would require further adaptation and iteration. We are excited about potential applications of  TopoPoint on large-scale, real-world datasets to further validate and refine its capabilities
>
> **If you have any further questions or suggestions about our article, we'd love to discuss our content with you.**

---

> ### Author Response · Authors · 2025-08-08
> **Sincere Request for Further Discussions**
>
> Dear Reviewer VkTs,
>
> Thanks again for your great efforts and constructive advice in reviewing this paper! With the discussion period drawing to a close, we expect your feedback and thoughts on our reply. We sincerely hope you can consider our reply in your assessment.
>
> We look forward to hearing from you. If you have any further questions, we would be more than happy to discuss with you. If you are satisfied with our response, we kindly hope you would consider increasing our score.
>
> Best Regards,
>
> Authors of Paper 1456

---

### Note · Authors · 2025-08-14

Dear Reviewers and ACs,

We would like to express our gratitude to all the reviewers and ACs for their valuable time and selfless dedication on our submission.

We are encouraged by the positive comments on the **well-motivated** (vKTs, gBnP, HgsV), **novel** (VkTs, HgsV), **well-written** (VkTs, P95Y, HgsV),  and **significant improvement** (gBnP, P95Y, HgsV).

We have made every effort to address all concerns raised by the reviewers in our rebuttal and subsequent discussions. In particular:

- For Reviewer ***VkTs***, we discussed open questions regarding the model’s **data sufficiency and inherent limitations hypothesis** in the context of future practical applications.
- For Reviewer ***gBnP***, we provided detailed responses to questions about the **resolution strategy** and comparisons with **TopoFormer**, supported by additional experimental results. Following the reviewer’s suggestion, we also verified that the PLGM module, when used alone as a **post-processing component**, can still improve the performance of other baselines. This broadens the applicability of our method and enriches the paper.
- For Reviewer ***P95Y***, we supplemented runtime benchmarks for TopoPoint and responsed concerns regarding **real-time deployment** as well as usability across different **weather conditions and regions**.
- For Reviewer ***HgsV***, we demonstrated through resource evaluations that **USGN** does not introduce significant overhead. We also discussed **interactions between subtasks** from three perspectives: logic, model, and experiments. Furthermore, we added experiments on **camera calibration** and conducted in-depth discussion of the model’s robustness, and addressed concerns about improvements for **downstream planning tasks**.

During the discussion phase, several reviewers noted that our responses **addressed their concerns** and expressed support for a positive rating, for which we are deeply appreciative.

Meanwhile, following the reviewers' suggestions, we are revising the manuscript to satisfy the high standards of the NeurIPS community. We sincerely hope that these rebuttal and comments can help the reviewers finalize the assessment of our work.

We finally thank you again can support our work during the successive stages of discussions and we promise to public all the codes of this work if accepted.

Best regards,

Authors of Paper 1456

---

### Decision · Program_Chairs · 2025-09-17

**Decision:**

Accept (poster)

**Comment:**

The paper focuses on lane topology prediction. It identifies issues in prior work where lanes with the same end point end up with divergent end points, and proposes a novel approach to jointly predicting lanes and end points.

Reviewers agree on the novelty of the approach and the strong results. Questions raised about experimental protocol were addressed by the authors in the rebuttal. As such, I recommend acceptance.